# Theoretical framework for mixed-potential-driven catalysis

Mo Yan [1], Nuning Anugrah Putri Namari[1], Junji Nakamura [2,3,4] ✉ & Kotaro Takeyasu [2,3,5] ✉

Mixed-potential-driven catalysis is expected to be a distinctive heterogeneous catalytic reaction that produces products different from those produced by thermal catalytic reactions without the application of external energy. Electrochemically, the mechanism is similar to that of corrosion. However, a theory that incorporates catalytic activity as a parameter has not been established. Herein, we report the theoretical framework of mixed-potential-driven catalysis, including exchange currents, as a parameter of catalytic activity. The mixed potential and partitioning of the overpotential were determined from the exchange current by applying the Butler–Volmer equation at a steady state far from equilibrium. Mixed-potential-driven catalysis is expected to open new areas not only in the concept of catalyst development but also in the field of energetics of biological enzymatic reactions.

Heterogeneous catalysis is crucial for solving various problems related to environment, energy, biology, and materials[1–4]. Generally, heterogeneous catalysis occurs thermally or electrochemically[5–7]. Recently, it has been suggested that thermal heterogeneous catalysis indeed includes electrochemical processes, leading to markedly different selectivity compared to conventional thermocatalysis[8–10]. In particular, electrode reactions that form mixed potentials typified by corrosion phenomena have attracted attention. Alternatively, the anodic and cathodic half-reactions occur in pairs on a single catalyst surface, where a mixed potential is expected to form if the catalyst is electrically conductive and a suitable electrolyte is present near the active sites, as shown in Fig. 1a. Here, we introduce the concept of "mixed-potential-driven catalysis" as such catalytic systems. The characteristic point of mixed-potential-driven catalysis is that anode and cathode catalysts are exposed to identical reactants, diverging from conventional electric cells where distinct reactants are supplied to each electrode. Intriguingly, it has been reported that some heterogeneous catalytic reactions of gas molecules involve mixed-potential-driven catalysis[9–16]. For example, it has been reported that $H_2O_2$ is selectively produced on various monometallic and bimetallic catalysts, which is considered to consist of an anodic reaction $H_2 \rightarrow 2H^+ + 2e^-$ and a cathodic reaction $O_2 + 2H^+ + 2e^- \rightarrow H_2O_2$[9]. Mixed-potential-driven catalysis has also been suggested for the oxidation of formic acid[10] and hydroquinone[11]. The occurrence of a mixed-potential-driven reaction during 4-nitrophenol hydrogenation was also proposed previously[15]. More interestingly, the mixed-potential-driven mechanism is caused by binary heterogeneous catalysts. The oxidation of alcohols

(hydroxymethylfurfural) on Au-Pd binary catalysts seems to proceed via mixed-potential-driven catalysis[12,13,16]. It is also worth noting that ethanol is produced with surprisingly high selectivity by $CO_2$ hydrogenation on CuPd binary powder catalysts in the presence of water, which is an unexpected product in thermal catalysis[14]. These reports strongly suggest that electrochemical processes play a role in controlling the activity and selectivity of heterogeneous catalysis without the need for external energy. Mixed-potential-driven catalysis is expected to open up a new category of heterogeneous catalysis in both basic research and industrial applications. However, the determining principle behind both the activity and selectivity, specifically the partitioning of the driving force for each half-reaction, has not been considered.

Mixed-potential-driven reactions have been mainly discussed in the field of electrochemistry, but not in heterogeneous catalysis. The mixed potential theory was first introduced by Wagner and Traud in 1938 in corrosion science[17]. As shown in Fig. 1b, the basic principle can be understood in terms of the polarization curves of two electrochemical reactions described by the Butler–Volmer equation, where $i_1, i_2$ and $\phi_1^{eq}, \phi_2^{eq}$ represent the current and equilibrium potential of two redox reactions $R_1 \rightleftarrows O_1 + e^-$ and $O_2 + e^- \rightleftarrows R_2$, with $\phi_2^{eq} > \phi_1^{eq}$. When the two reactions proceed concurrently, $i_1 + i_2 = 0$ is satisfied owing to the conservation of electric charge forming a mixed-potential $\phi^{mix}$. Here, $\phi^{mix} - \phi_1^{eq}$ and $\phi^{mix} - \phi_2^{eq}$ act as the overpotentials $\eta_1^{mix}$ and $\eta_2^{mix}$ in the two reactions[18]. In the literature[19–33], mixed potentials and reaction currents have been formulated for simple pair and parallel reactions with the effects of mass diffusion. Notably, an

[1]Graduate School of Science and Technology, University of Tsukuba, 1-1-1 Tennodai, Tsukuba, Ibaraki 305-8573, Japan. [2]Department of Materials Science, Faculty of Pure and Applied Sciences, University of Tsukuba, 1-1-1 Tennodai, Tsukuba, Ibaraki 305-8573, Japan. [3]Tsukuba Research Centre for Energy and Materials Science, University of Tsukuba, 1-1-1 Tennodai, Tsukuba, Ibaraki 305-8573, Japan. [4]International Institute for Carbon-Neutral Energy Research (I²CNER), Kyushu University, 744 Motooka, Nishi-ku, Fukuoka-shi, Fukuoka 819-0395, Japan. [5]R&D Center for Zero CO2 Emission with Functional Materials, University of Tsukuba, 1-1-1 Tennodai, Tsukuba, Ibaraki 305-8573, Japan. ✉e-mail: nakamura.junji.700@m.kyushu-u.ac.jp; takeyasu.kotaro.gt@u.tsukuba.ac.jp

**Fig. 1 | A typical mixed-potential-driven reaction.**
**a** Electrons released in the oxidation reaction from reductant $R_1$ to oxidant $O_1$ are used in the reduction reaction from oxidant $O_2$ to reductant $R_2$.
**b** Illustrative polarization curves for the cathodic and anodic half-reactions. The mixed potential is the point at which the net of the cathodic and anodic currents is zero.

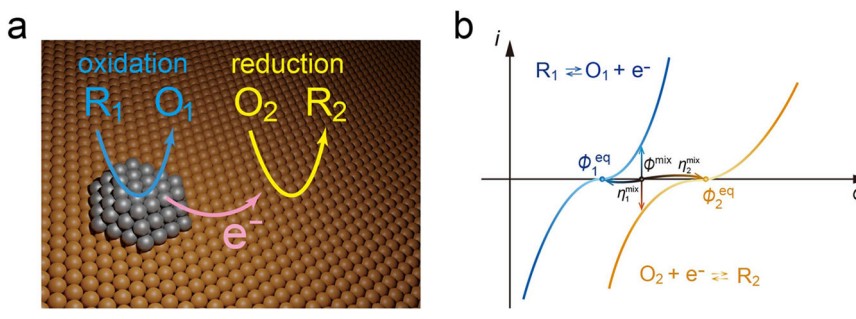

**Fig. 2 | A proposed mixed-potential-driven catalysis model. a** Schematic of a mixed-potential-driven catalytic reaction occurring on the catalyst composed of component I and II. Cathodic and anodic half-reactions can occur in each of the component I and II. Electrons are transferred within and between the component I and II. **b** Illustration of the four polarization curves for the cathodic and anodic half-reactions on catalyst component I and II. The mixed potential is the point at which the sum of the four currents is zero.

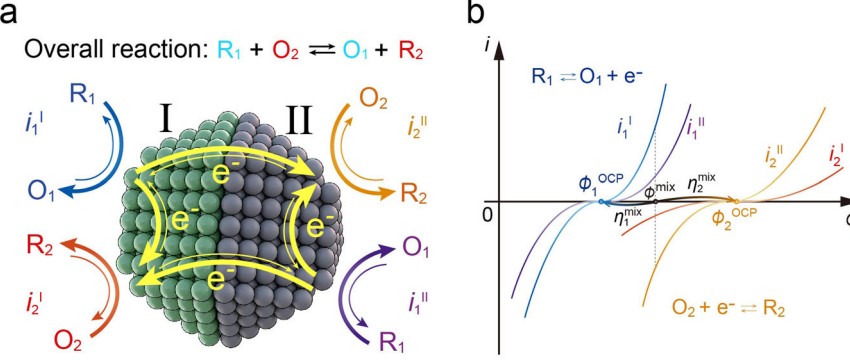

overpotential accelerates the electrochemical reactions[33,34]. Therefore, the partitioning of the overpotential is essential because some kinetically unfavorable half-reactions can be accelerated with a high overpotential by coupling a kinetically favorable half-reaction. It is argued that to achieve the same rate, a larger overpotential is required to conduct the electrode reaction with a higher activation barrier[9]. Despite advances in the understanding of mixed-potential-driven catalysis, determining the overpotential as the driving force based on the catalytic activity has not been elucidated so far.

Mixed-potential-driven catalysis is classified as a non-equilibrium thermodynamic phenomenon. The chemical potential drop between the reactants and products becomes the driving force for the reaction, overcoming the activation energy and converting it into energy to increase the reaction rate[35]. When the equilibrium state is achieved, the driving force becomes zero, which is converted to the heat of reaction[36]. Prigogine constructed a theoretical framework based on entropy change to conserve the energy[37,38]. However, it has not been explicitly stated that $d_iS$ (entropy production) corresponds to the overpotentials that promote the reaction in the case of mixed-potential-driven catalysis. In this paper, we extend Prigogine's theory to the mixed-potential-driven catalysis and present the kinetic equations. Enzymatic reaction systems in living organisms, such as glucose oxidase and lactate oxidase, may also proceed via a mixed-potential-driven reaction, in which the anodic and cathodic reactions are paired[39–43]. Thus, the framework of mixed-potential-driven catalysis is fundamental for considering the energy pathways of how entropy is generated in the body, which is used to drive metabolic reactions, maintain body temperature as heat, and dissipated outside. The non-equilibrium theory of mixed-potential-driven catalysis is expected to improve our understanding of the energetics of biological systems.

In this study, we present an equation for the conversion of the Gibbs free energy drop between the cathodic and anodic half-reactions into overpotentials by the formation of a mixed potential. In particular, the equation explains how the catalytic activity plays a pivotal role in determining the mixed potential, overpotentials, reaction current, and selection of the cathodic and anodic reactions. This is the equation for the concept of mixed-potential-driven catalysis. This concept is important in the

development of kinetically difficult catalytic reactions, understanding the energy transfer of enzymatic reactions in living organisms, and in the non-equilibrium theory of chemical reactions.

## Results

### Driving force of mixed-potential-driven catalysis

First, we show how the total driving force of the entire mixed-potential-driven catalytic reaction system is distributed to the overpotentials for accelerating the anodic and cathodic half-reactions, depending on the catalytic activity of the catalysts. We consider a mixed potential system, as shown in Fig. 2a, where we assume one-electron transfer processes of anodic reaction 1 and cathodic reaction 2 occurring at both components I and II of the catalyst.

$$R_1 \rightleftharpoons O_1 + e^- \qquad (1)$$

$$O_2 + e^- \rightleftharpoons R_2 \qquad (2)$$

The net reaction is expressed by the following equation.

$$R_1 + O_2 \rightleftharpoons O_1 + R_2 \qquad (3)$$

Electrochemically, microelectrodes I and II can be regarded as short-circuited, with both electrodes exposed to identical gas or liquid conditions, regardless of whether they are spatially separated. Unlike ordinary electrochemical cells, the distinction between the anode and cathode is not fixed before starting the reaction. Consequently, Eqs. (1) and (2) each occur on two different catalyst components leading to $i_1^I$, $i_2^I$, $i_1^{II}$, and $i_2^{II}$, where one assumes that equilibrium potential of reaction 1 ($\phi_1^{eq}$) is lower than that of reaction 2 ($\phi_2^{eq}$). The potential difference of $\phi_2^{eq} - \phi_1^{eq}$ corresponds to the total driving force of the net reaction Eq. (3).

To estimate the mixed potential and current at the mixed potential, it is necessary to analyze the polarization curve, which depends on the catalytic activity and is expressed by the Butler–Volmer equation. The currents of the

electrochemical half-reactions (1) and (2) on components I and II are given by the Butler–Volmer equation with no mass-transfer effect:

$$i_1^{\text{I}} = i_1^{\text{I}\,0}\left(e^{(1-\alpha_1)f\left(\phi-\phi_1^{\text{eq}}\right)} - e^{-\alpha_1 f\left(\phi-\phi_1^{\text{eq}}\right)}\right) \tag{4}$$

$$i_2^{\text{I}} = i_2^{\text{I}\,0}\left(e^{(1-\alpha_2)f\left(\phi-\phi_2^{\text{eq}}\right)} - e^{-\alpha_2 f\left(\phi-\phi_2^{\text{eq}}\right)}\right) \tag{5}$$

$$i_1^{\text{II}} = i_1^{\text{II}\,0}\left(e^{(1-\alpha_1)f\left(\phi-\phi_1^{\text{eq}}\right)} - e^{-\alpha_1 f\left(\phi-\phi_1^{\text{eq}}\right)}\right) \tag{6}$$

$$i_2^{\text{II}} = i_2^{\text{II}\,0}\left(e^{(1-\alpha_2)f\left(\phi-\phi_2^{\text{eq}}\right)} - e^{-\alpha_2 f\left(\phi-\phi_2^{\text{eq}}\right)}\right) \tag{7}$$

where $f = F/RT$ and $F$, $R$, and $T$ are the Faraday constant, gas constant, and temperature, respectively. $\alpha_1$ and $\alpha_2$ are the transfer coefficients for reaction 1 and reaction 2, respectively. $i_1^{\text{I}\,0}$, $i_2^{\text{I}\,0}$, $i_1^{\text{II}\,0}$, and $i_2^{\text{II}\,0}$ are the exchange currents for reactions 1 and 2 on components I and II, respectively. $\phi - \phi_1^{\text{eq}}$ and $\phi - \phi_2^{\text{eq}}$ are the overpotentials $\eta_1$ and $\eta_2$ for reactions 1 and 2, respectively. The exchange current $i^0$ corresponds to the catalytic activity and determines the shape of the polarization curve[44]. Here, the mixed potential $\phi^{\text{mix}}$ is defined as the potential at which the net current is zero, as shown in Fig. 2b[17].

$$i_1^{\text{I}} + i_1^{\text{II}} + i_2^{\text{I}} + i_2^{\text{II}} = 0 \tag{8}$$

By substituting Eqs. (4)–(7) into Eq. (8), one can calculate the mixed potential $\phi^{\text{mix}}$ numerically using practical values of exchange currents, equilibrium potentials, and transfer coefficients. On the other hand, one can obtain the relationship among mixed potentials, overpotentials, and exchange currents based on analytical solutions with the assumption of identical transfer coefficients ($\alpha_1 = \alpha_2 = \alpha$). Then, one can derive Eq. (9) for $\phi^{\text{mix}}$ (detailed derivation shown in Supplementary Note 1).

$$\phi^{\text{mix}} = \frac{1}{f} \ln \frac{\left(i_1^{\text{I}\,0} + i_1^{\text{II}\,0}\right)e^{\alpha f \phi_1^{\text{eq}}} + \left(i_2^{\text{I}\,0} + i_2^{\text{II}\,0}\right)e^{\alpha f \phi_2^{\text{eq}}}}{\left(i_1^{\text{I}\,0} + i_1^{\text{II}\,0}\right)e^{-(1-\alpha)f \phi_1^{\text{eq}}} + \left(i_2^{\text{I}\,0} + i_2^{\text{II}\,0}\right)e^{-(1-\alpha)f \phi_2^{\text{eq}}}} \tag{9}$$

The absolute value of the anodic and cathodic currents must be the same, which is the current at the mixed potential ($i^{\text{mix}}$) for the net reaction.

$$i^{\text{mix}} = \left|i_1^{\text{I}} + i_1^{\text{II}}\right| = \left|i_2^{\text{I}} + i_2^{\text{II}}\right| \tag{10}$$

Substituting Eq. (9) into Eqs. (4), (5), and (10) gives $i^{\text{mix}}$.

$$
\begin{aligned}
i^{\text{mix}} &= \left(i_1^{\text{I}\,0} + i_1^{\text{II}\,0}\right)\left(e^{(1-\alpha)f\left(\phi^{\text{mix}}-\phi_1^{\text{eq}}\right)} - e^{-\alpha f\left(\phi^{\text{mix}}-\phi_1^{\text{eq}}\right)}\right) \\
&= \left(i_1^{\text{I}\,0} + i_1^{\text{II}\,0}\right)\left[\left(\frac{\left(i_1^{\text{I}\,0} + i_1^{\text{II}\,0}\right) + \left(i_2^{\text{I}\,0} + i_2^{\text{II}\,0}\right)e^{\alpha f\left(\phi_2^{\text{eq}}-\phi_1^{\text{eq}}\right)}}{\left(i_1^{\text{I}\,0} + i_1^{\text{II}\,0}\right) + \left(i_2^{\text{I}\,0} + i_2^{\text{II}\,0}\right)e^{-(1-\alpha)f\left(\phi_2^{\text{eq}}-\phi_1^{\text{eq}}\right)}}\right)^{1-\alpha} \right. \\
&\quad \left. - \left(\frac{\left(i_1^{\text{I}\,0} + i_1^{\text{II}\,0}\right) + \left(i_2^{\text{I}\,0} + i_2^{\text{II}\,0}\right)e^{\alpha f\left(\phi_2^{\text{eq}}-\phi_1^{\text{eq}}\right)}}{\left(i_1^{\text{I}\,0} + i_1^{\text{II}\,0}\right) + \left(i_2^{\text{I}\,0} + i_2^{\text{II}\,0}\right)e^{-(1-\alpha)f\left(\phi_2^{\text{eq}}-\phi_1^{\text{eq}}\right)}}\right)^{-\alpha}\right]
\end{aligned}
\tag{11}
$$

It is shown that $i^{\text{mix}}$ is a function of the driving force $\phi_2^{\text{eq}} - \phi_1^{\text{eq}}$ and the exchange current for each reaction. When we define $\phi^{\text{mix}} - \phi_1^{\text{eq}}$ and $\phi^{\text{mix}} - \phi_2^{\text{eq}}$ as $\eta_1^{\text{mix}}$ and $\eta_2^{\text{mix}}$, respectively, $\phi_2^{\text{eq}} - \phi_1^{\text{eq}}$ can be regarded as the sum of overpotentials $|\eta_1^{\text{mix}}| + |\eta_2^{\text{mix}}|$ to promote catalytic reactions as applied from the outside. Here, the partitioning of $\phi_2^{\text{eq}} - \phi_1^{\text{eq}}$ to the overpotentials $|\eta_1^{\text{mix}}|$

and $|\eta_2^{\text{mix}}|$ can be expressed using Eq. (9).

$$\left|\eta_1^{\text{mix}}\right| = \phi^{\text{mix}} - \phi_1^{\text{eq}} = \frac{1}{f}\ln \frac{e^{\alpha f\left(\phi_2^{\text{eq}}-\phi_1^{\text{eq}}\right)} + \frac{\left(i_1^{\text{I}\,0}+i_1^{\text{II}\,0}\right)}{\left(i_2^{\text{I}\,0}+i_2^{\text{II}\,0}\right)}}{e^{-(1-\alpha)f\left(\phi_2^{\text{eq}}-\phi_1^{\text{eq}}\right)} + \frac{\left(i_1^{\text{I}\,0}+i_1^{\text{II}\,0}\right)}{\left(i_2^{\text{I}\,0}+i_2^{\text{II}\,0}\right)}} \tag{12}$$

$$\left|\eta_2^{\text{mix}}\right| = \phi_2^{\text{eq}} - \phi^{\text{mix}} = \frac{1}{f}\ln \frac{e^{(1-\alpha)f\left(\phi_2^{\text{eq}}-\phi_1^{\text{eq}}\right)} + \frac{\left(i_2^{\text{I}\,0}+i_2^{\text{II}\,0}\right)}{\left(i_1^{\text{I}\,0}+i_1^{\text{II}\,0}\right)}}{e^{-\alpha f\left(\phi_2^{\text{eq}}-\phi_1^{\text{eq}}\right)} + \frac{\left(i_2^{\text{I}\,0}+i_2^{\text{II}\,0}\right)}{\left(i_1^{\text{I}\,0}+i_1^{\text{II}\,0}\right)}} \tag{13}$$

It should be noted that in Eqs. (12) and (13) the total overpotential $\phi_2^{\text{eq}} - \phi_1^{\text{eq}}$ is partitioned to $|\eta_1^{\text{mix}}|$ and $|\eta_2^{\text{mix}}|$ according to the exchange current or catalytic activity. The crucial factor influencing the overpotential $|\eta_1^{\text{mix}}|$ and $|\eta_2^{\text{mix}}|$ is the ratio of $\left(i_1^{\text{I}\,0} + i_1^{\text{II}\,0}\right) : \left(i_2^{\text{I}\,0} + i_2^{\text{II}\,0}\right)$.

Assuming that a single oxidation reaction and a single reduction reaction take place on each catalyst component (while $i_1^{\text{I}\,0}$ and $i_2^{\text{II}\,0}$ remain, but $i_2^{\text{I}\,0}$ and $i_1^{\text{II}\,0}$ are zero), it is clearly shown in Eqs. (S2–2) and (S2–3) that the ratio of $i_1^{\text{I}\,0} : i_2^{\text{II}\,0}$ determines the overpotential $|\eta_1^{\text{mix}}|$ and $|\eta_2^{\text{mix}}|$ (detailed discussion shown in Supplementary Note 2 and Supplementary Fig. 1). For example, if $i_1^{\text{I}\,0} \ll i_2^{\text{II}\,0}$, $|\eta_1^{\text{mix}}| \gg |\eta_2^{\text{mix}}|$ will be obtained, $|\eta_1^{\text{mix}}|$ and $|\eta_2^{\text{mix}}|$ will approach $\phi_2^{\text{eq}} - \phi_1^{\text{eq}}$ and zero, respectively. This example is significant for heterogeneous catalysis because the catalytically difficult reaction 1 with small $i_1^{\text{I}\,0}$ can be promoted by applying larger overpotential of $|\eta_1^{\text{mix}}|$ that corresponds to the total driving force $\phi_2^{\text{eq}} - \phi_1^{\text{eq}}$ of the net reaction.

## Overpotential partitioning depending on exchange current

To comprehend the physical meaning of overpotential partitioning, which is the relationship between the overpotential and exchange current, two approximation methods were adopted. One is the linear approximation of the Taylor expansion for small overpotentials, and the other is Tafel approximation for large overpotentials (see Supplementary Note 1 for the case of Tafel approximation; the error estimation is discussed in Supplementary Note 4, Supplementary Fig. 2, and Supplementary Table 1). Here, in the linear approximation, for the catalyst component I, the currents in Eqs. (4) and (5) are approximated as follows:

$$i_1^{\text{I}} = i_1^{\text{I}\,0}f(\phi - \phi_1^{\text{eq}}) \tag{14}$$

$$i_2^{\text{I}} = i_2^{\text{I}\,0}f(\phi - \phi_2^{\text{eq}}) \tag{15}$$

For the currents on catalyst component II, "I" in Eqs. (14) and (15) can be replaced by "II". Combining the four equations for currents with Eq. (8) yields $\phi^{\text{mix}}$.

$$\phi^{\text{mix}} = \frac{\left(i_1^{\text{I}\,0} + i_1^{\text{II}\,0}\right)\phi_1^{\text{eq}} + \left(i_2^{\text{I}\,0} + i_2^{\text{II}\,0}\right)\phi_2^{\text{eq}}}{i_1^{\text{I}\,0} + i_1^{\text{II}\,0} + i_2^{\text{I}\,0} + i_2^{\text{II}\,0}} \tag{16}$$

Equation (16) clearly shows that the mixed potential is determined by internal division with a ratio of $(i_1^{\text{I}\,0} + i_1^{\text{II}\,0}) : (i_2^{\text{I}\,0} + i_2^{\text{II}\,0})$. Simultaneously, the current at the mixed potential is obtained as follows:

$$i^{\text{mix}} = \left(i_1^{\text{I}\,0} + i_1^{\text{II}\,0}\right)f(\phi^{\text{mix}} - \phi_1^{\text{eq}}) = \frac{\left(i_1^{\text{I}\,0} + i_1^{\text{II}\,0}\right)\left(i_2^{\text{I}\,0} + i_2^{\text{II}\,0}\right)}{i_1^{\text{I}\,0} + i_1^{\text{II}\,0} + i_2^{\text{I}\,0} + i_2^{\text{II}\,0}}\left(\phi_2^{\text{eq}} - \phi_1^{\text{eq}}\right) \tag{17}$$

$i^{\text{mix}}$ corresponds to the apparent catalytic activity in the mixed-potential-driven catalysis, which is determined by exchange current and the driving force of $\phi_2^{\text{eq}} - \phi_1^{\text{eq}}$.

In addition, the overpotentials $|\eta_1^{mix}|$ and $|\eta_2^{mix}|$ are rewritten using $\phi^{mix}$ of Eq. (16).

$$|\eta_1^{mix}| = \phi^{mix} - \phi_1^{eq} = \frac{i_2^{I0} + i_2^{II0}}{i_1^{I0} + i_1^{II0} + i_2^{I0} + i_2^{II0}}(\phi_2^{eq} - \phi_1^{eq}) \quad (18)$$

$$|\eta_2^{mix}| = \phi_2^{eq} - \phi^{mix} = \frac{i_1^{I0} + i_1^{II0}}{i_1^{I0} + i_1^{II0} + i_2^{I0} + i_2^{II0}}(\phi_2^{eq} - \phi_1^{eq}) \quad (19)$$

Then, the ratio of the overpotential is expressed by:

$$|\eta_1^{mix}| : |\eta_2^{mix}| = \frac{i_2^{I0} + i_2^{II0}}{i_1^{I0} + i_1^{II0} + i_2^{I0} + i_2^{II0}}(\phi_2^{eq} - \phi_1^{eq})$$
$$: \frac{i_1^{I0} + i_1^{II0}}{i_1^{I0} + i_1^{II0} + i_2^{I0} + i_2^{II0}}(\phi_2^{eq} - \phi_1^{eq}) = \frac{1}{i_1^{I0} + i_1^{II0}} : \frac{1}{i_2^{I0} + i_2^{II0}} \quad (20)$$

Here, it is clear that the driving force of the entire reaction, $\phi_2^{eq} - \phi_1^{eq}$, is partitioned to overpotential $|\eta_1^{mix}|$ and $|\eta_2^{mix}|$ according to the ratio of the sum of the exchange current $i_1^{I0} + i_1^{II0}$ and $i_2^{I0} + i_2^{II0}$ for reactions 1 and 2, i.e., the catalytic activity. Figure 3 is a conceptual electric series circuit representing a mixed-potential-driven catalytic reaction where $\phi_2^{eq} - \phi_1^{eq}$ corresponds to the electromotive source due to reaction 1 and 2, and $|\eta_1^{mix}|$ and $|\eta_2^{mix}|$ corresponds to the overpotentials of reaction 1 and 2 without external electric work.

Here, $r_1$ and $r_2$ are the so-called charge-transfer resistances depending on the catalytic activity, which are proportional to $1/(i_1^{I0} + i_1^{II0})$ and $1/(i_2^{I0} + i_2^{II0})$ for the half-reactions 1 and 2, respectively[28,44]. The

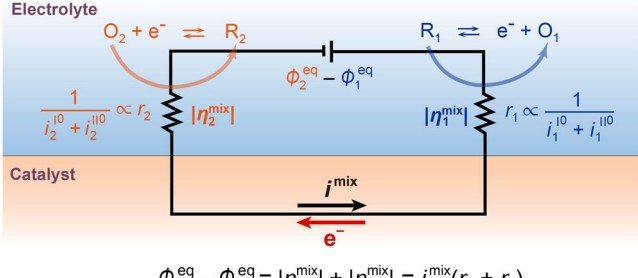

**Overall reaction: R₁ + O₂ → O₁ + R₂**

**Fig. 3 | A conceptual electric series circuit illustrates a mixed-potential-driven catalytic reaction.** The internal total voltage $\phi_2^{eq} - \phi_1^{eq}$ is due to the Gibbs free energy drop $-\Delta G_r$ across the entire mixed-potential-driven catalytic reaction. The charge-transfer resistance ($r_1$ and $r_2$), proportional to the reciprocal of the exchange current, plays a role similar to electrical resistors in a circuit. The voltage drops, $i^{mix}r_1$ and $i^{mix}r_2$, signifies the overpotentials $|\eta_1^{mix}|$ and $|\eta_2^{mix}|$, following the voltage divider rule. The energy utilized for driving reactions 1 and 2 eventually transforms into Joule heat ($\eta i$).

partitioning of the driving force $\phi_2^{eq} - \phi_1^{eq}$ into $|\eta_1^{mix}|$ and $|\eta_2^{mix}|$ in the mixed-potential-driven catalysis follows the voltage divider rule in the series circuit as $i^{mix}r_1$ and $i^{mix}r_2$. This implies that a larger overpotential is partitioned to accelerate processes with a higher charge-transfer resistance.

One can regard this as short circuit where no external work and the Gibbs free energy term is converted to the overpotentials of $|\eta_1^{mix}|$ and $|\eta_2^{mix}|$ away from equilibrium. Here, the driving force $\phi_2^{eq} - \phi_1^{eq}$ corresponds to the Gibbs free energy change $\Delta G_r$ of the net reaction with certain concentration of molecules involved at which the reaction proceeds[32,45,46].

$$-\Delta G_r = F(\phi_2^{eq} - \phi_1^{eq}) = F(|\eta_1^{mix}| + |\eta_2^{mix}|) \quad (21)$$

We assumed that half-reactions 1 and 2 are one-electron transfer reactions so that the number of "moles of electrons" exchanged in the half-reactions, $n$, is equal to 1 and drops away in Eq. (21). This mechanism efficiently drives reactions by utilizing the overpotential to accelerate the forward reaction and decelerate the backward reaction[46]. This differs from thermocatalytic reactions, which use a driving force to accelerate both the forward and backward reactions. This is a non-equilibrium steady state, which is discussed in detail below. However, it is noted here that the energy used to drive reactions 1 and 2 will be dissipated as Joule heat expressed as:

$$\eta_1^{mix}i_1^I : \eta_1^{mix}i_1^{II} : \eta_2^{mix}i_2^I : \eta_2^{mix}i_2^{II}$$
$$= \eta_1^{mix} \times i_1^{I0}f\eta_1^{mix} : \eta_1^{mix} \times i_1^{II0}f\eta_1^{mix} : \eta_2^{mix} \times i_2^{I0}f\eta_2^{mix} : \eta_2^{mix} \times i_2^{II0}f\eta_2^{mix}$$
$$= \frac{i_1^{I0}}{(i_1^{I0} + i_1^{II0})^2} : \frac{i_1^{II0}}{(i_1^{I0} + i_1^{II0})^2} : \frac{i_2^{I0}}{(i_2^{I0} + i_2^{II0})^2} : \frac{i_2^{II0}}{(i_2^{I0} + i_2^{II0})^2} \quad (22)$$

Equation (22) indicates that the heat generated by each reaction was determined by the exchange current. By contrast, thermochemical reactions directly convert the Gibbs free energy to heat. This distinction is one of the secrets to how mixed-potential-driven catalysis can efficiently accelerate reactions.

## Direction of the current flow or electron transfer

In mixed-potential-driven catalysis, the direction of current flow or electron transfer is governed by the exchange current or catalytic activity. Understanding how electrons are transferred between the components is crucial for catalyst design. However, before starting the reaction, the anode and cathode components are unknown. After initiation of the reaction, the magnitude of the exchange current or catalytic activity determines the direction of the current flow or electron transfer and distinguishes between the anode and cathode. Essentially, the roles of components I and II are uncertain and interchangeable.

This uncertainty leads to three possible cases regarding the direction of the current flow or the designation of components I and II as the anode and cathode of the catalyst, respectively, as shown in Fig. 4. Case (a): Overall, the anodic and cathodic current predominates in component I and II, respectively. Case (b): The cathodic and anodic current predominate in

**Fig. 4 | Direction of the current flow or electron transfer on the catalyst between component I and II occurs through three possible cases.**
**a** Component I is anode and component II is cathode, i.e., the current flows from II to I; (**b**) Component I is cathode and component II is anode, i.e., the current flows from I to II; (**c**) The current flows within both components I and II but there is no current flows between I and II.

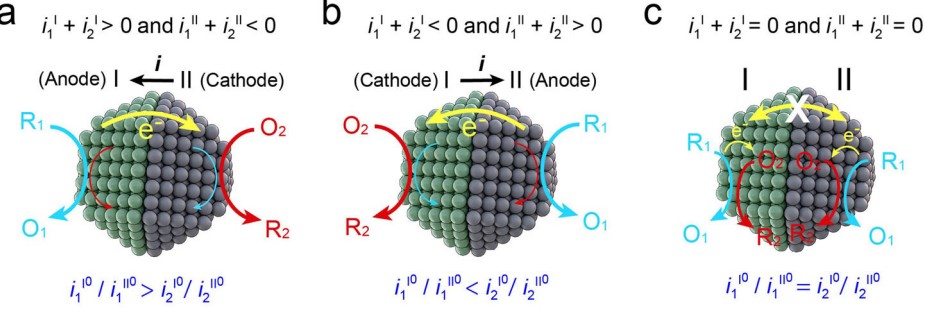

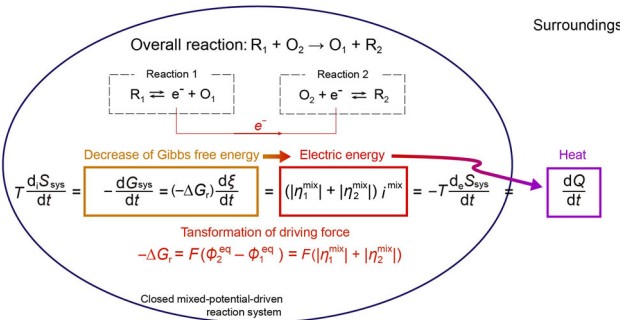

**Fig. 5 | Steady-state mixed-potential-driven catalysis occurs in a closed, iso-thermal, and isobaric system.** Surroundings are enclosed by rigid adiabatic walls, completely isolated from the external world, a common experimental approximation. At steady-state, the "internal" entropy created in the reaction system $(T d_i S_{sys}/dt)$ which exactly balances the "exchange" entropy to the surroundings $(-T d_e S_{sys}/dt)$, and would be dissipated as heat $(dQ/dt)$ in the surroundings. At any particular time, in the mixed-potential-driven catalysis, Gibbs free energy drop of the net reaction $(-\Delta G_r)$ undergoes transformation into the overpotentials $(|\eta_1^{mix}|$ and $|\eta_2^{mix}|)$, which serve to accelerate each of half-reactions, and are ultimately dissipated as Joule heat to the surroundings through the exchange entropy.

component I and II, respectively. Case (c): Anodic and cathodic currents proceed in pairs in components I and II, respectively, resulting in no current flow between them. By substituting the approximation equations (both the Tafel and linear approximation methods yielded identical results) for the currents of reactions 1 and 2 on components I and II, the direction of the current flow in the three cases can be expressed by the exchange currents as follows (detailed derivation shown in Supplementary Note 3):

$$Case\ A: \frac{i_1^{I\,0}}{i_1^{II\,0}} > \frac{i_2^{I\,0}}{i_2^{II\,0}}\ or\ \frac{i_1^{I\,0}}{i_2^{I\,0}} > \frac{i_1^{II\,0}}{i_2^{II\,0}} \tag{23}$$

$$Case\ B: \frac{i_1^{I\,0}}{i_1^{II\,0}} < \frac{i_2^{I\,0}}{i_2^{II\,0}}\ or\ \frac{i_1^{I\,0}}{i_2^{I\,0}} < \frac{i_1^{II\,0}}{i_2^{II\,0}} \tag{24}$$

$$Case\ C: \frac{i_1^{I\,0}}{i_1^{II\,0}} = \frac{i_2^{I\,0}}{i_2^{II\,0}}\ or\ \frac{i_1^{I\,0}}{i_2^{I\,0}} = \frac{i_1^{II\,0}}{i_2^{II\,0}} \tag{25}$$

Equations (23)–(25) indicate that the current flow direction is kinetically governed by the exchange current ratio or catalytic activity. The exchange current values are sensitive to substance concentrations and pH, as reported in the literature[47]. Controlling the current direction by adjusting the exchange current can help researchers harness the benefits of the internal electric field of the catalyst and enhance selectivity for the desired products.

## Non-equilibrium thermodynamics for mixed-potential-driven catalysis at steady-state

Herein, non-equilibrium thermodynamics at steady-state are discussed for the mixed-potential-driven catalysis based on the entropy production concept proposed by Prigogine. The starting point of Prigogine's theory is to express the changes in entropy as the sum of two parts:

$$dS_{sys} = d_e S_{sys} + d_i S_{sys} \tag{26}$$

where $dS_{sys}$ is the total variation in the entropy of a system, $d_e S_{sys}$ is the entropy change of the system owing to the exchange of matter and energy with the exterior, and $d_i S_{sys}$ is the entropy produced by the irreversible

processes inside the system[38]. The entropy production term $d_i S_{sys}$, can serve as a basis for the systematic description of irreversible processes occurring in a system, and $d_i S_{sys}$ is always non-negative. Moreover, in the steady-state, the time derivative of the system entropy, $dS_{sys}/dt$, is zero, that is, the entropy spontaneously generated inside the system is balanced by a flow of the entropy exchange with the outside[38,48]:

$$\frac{dS_{sys}}{dt} = \frac{d_e S_{sys}}{dt} + \frac{d_i S_{sys}}{dt} = 0\ or\ \frac{d_i S_{sys}}{dt} = -\frac{d_e S_{sys}}{dt} \tag{27}$$

For chemical processes in a closed system at constant pressure and temperature, the rate of entropy production can be expressed in the form of the Gibbs free energy[38]:

$$T\frac{d_i S_{sys}}{dt} = -\frac{dG_{sys}}{dt} = -\frac{dG_{sys}}{d\xi}\frac{d\xi}{dt} = (-\Delta G_r)\frac{d\xi}{dt} \tag{28}$$

where $dG_{sys}$ is the change of total Gibbs free energy of the reaction system, $\xi$ is the extent of reaction, $d\xi/dt$ is the rate of the reaction, and $-\Delta G_r$ is the driving force for the net reaction corresponding to affinity $A$ in Prigogine's textbook (defined as $-dG_{sys}/d\xi$ and shown in Supplementary Fig. 3). In electrical conduction system, the rate of entropy production corresponds to the Joule heat (per unit time):

$$T\frac{d_i S_{sys}}{dt} = VI = \frac{dQ'}{dt} \tag{29}$$

where $V$ is the potential difference across the entire conductor, $I$ is the convention electric current, and $dQ'$ is the Joule heat generated from the electric current[38,49,50].

The equations above are generally present in textbook. Applying these equations to the mixed-potential-driven catalysis allows us to describe the energy conversion pathway within the framework of non-equilibrium thermodynamics at steady-state, as follows (detailed derivation shown in Supplementary Note 5):

$$T\frac{d_i S_{sys}}{dt} = -\frac{dG_{sys}}{dt} = (-\Delta G_r)\frac{d\xi}{dt} = F(\phi_2^{eq} - \phi_1^{eq})\frac{d\xi}{dt}$$
$$= (|\eta_1^{mix}| + |\eta_2^{mix}|)i^{mix} = -T\frac{d_e S_{sys}}{dt} = \frac{dQ}{dt} \tag{30}$$

where $dQ$ denotes the Joule heat generated due to the reaction. Equation (30) can be illustrated using a closed, isothermal, and isobaric mixed-potential-driven catalytic reaction system at steady-state, as depicted in Fig. 5. We may consider that the surroundings of the reaction system are enclosed by rigid adiabatic walls, meaning that the surroundings achieve equilibrium throughout; that is, the temperature, pressure, and chemical potentials remain constant[48]. Clearly, the mixed-potential-driven catalysis theory can be categorized as a non-equilibrium theory. Note here that the mixed-potential-driven catalysis provides a mechanism of internal driving force transformation where the Gibbs free energy drop of the net reaction $(-\Delta G_r)$ is converted to overpotentials for the two half-reactions $(|\eta_1^{mix}|$ and $|\eta_2^{mix}|)$ inside the reaction system. Thus, it can be concluded that the mixed-potential-driven catalysis converts the Gibbs free energy driving force to internal electric energy and finally to Joule heat.

## Discussion

Mixed-potential-driven catalysis occurs when the anodic and cathodic reactions are short-circuited in an appropriate electrolyte, and the difference in Gibbs free energy between the anodic and cathodic reactions converts into overpotentials to promote both reactions. In this study, we generalize the theory of mixed-potential-driven catalysis, including the parameters of catalytic activity. We formulate the relationship between the Gibbs free energy and the overpotential using the exchange current as the catalytic activity. The present theoretical analysis has clearly demonstrated how the

mixed potential is determined, overpotential is partitioned, and anode and cathode are selected by the exchange current. Although the present theoretical framework is fundamental and is constructed using a simple model, many additional effects must be taken into account for further development and application in the future.

In principle, the theoretical framework of mixed-potential-driven catalysis can be applied to both solid-gas and solid-liquid interfaces, where an electrolyte is necessary to convey ions. One open issue is how the overpotential is applied to electrode reactions at solid-gas and solid-liquid interfaces. At present, we consider that the overpotential in mixed-potential-driven catalysis corresponds to an electric double layer (EDL) at the catalyst surface, where electrochemical reactions are accelerated or deaccelerated. The nanoscale EDL at the interface may play a large role, where the shape of local electric field of EDL is determined by concentrations and distributions of cations, anions, and electrons depending on the overpotential. That local electric field should critically influence the reaction kinetics. Therefore, it is important to study the local structure of the EDL at the gas-solid and liquid-solid interfaces. Furthermore, as the size of the electrode decreases, a strong electric field may be generated. Thus, it is necessary to clarify the relationship among the overpotential, electrode structure, and EDL structure. Recent studies have reported that EDLs at spatially distant cathodes and anodes change in an intrinsically coupled manner[51]. Future research will employ both experimental and theoretical studies of the EDL in mixed-potential-driven catalysis.

The mass transport effect is not included in the present theoretical model because the main aim of this study was to show that catalytic activity mainly determines the mixed potential. However, it is necessary to consider the non-linear mass transport effect to determine the current value in addition to Bulter-Volmer equations. The position of mixed potential and reaction rate are shifted depending on the mass transport effect (Supplementary Fig. 5), as discussed in Supplementary Note 7. Even more complex, electron transfer numbers, transfer coefficients, and the co-occurrence of thermal reactions must be considered in the kinetic model of mixed-potential-driven catalysis. In actual catalytic reaction systems, these additional effects must be considered in an extremely complex manner. Therefore, it is necessary to combine research on relatively simple systems to approach real catalytic reactions that involve extremely complex elements.

Another important aspect of mixed-potential-driven catalysis is non-equilibrium thermodynamics. Mixed-potential-driven catalysis will be particularly important in the energetics of enzymatic reactions of biological systems (discussed in Supplementary Note 6 and Supplementary Fig. 4). As described above, the Gibbs free energy drop or uncompensated heat ($d_i S_{sys}$) is first converted into overpotential and then into heat. This energy conversion is a characteristic feature of non-equilibrium thermodynamics and is expected to greatly contribute to the future development of non-equilibrium thermodynamics itself. The energy conversion is particularly important in enzymatic reactions of biological systems is important because the mechanism of thermogenesis in biological systems is expected to be closely linked to the present non-equilibrium theory[52].

## Data availability

Data sharing is not applicable to this article as no datasets were generated or analyzed during the current study.

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

## Acknowledgements
This work was supported by JSPS Grant-in-Aid for Scientific Research (KAKENHI) Grant Number 23H05459, JST the establishment of university fellowships towards the creation of science technology innovation Grant Number JPMJFS2106, Project for University-Industry Cooperation Strengthening in Tsukuba, and TRiSTAR Program, a Top Runner Development Program Engaging Universities, National Labs, and Companies. M.Y., N.A.P.N., J.N., and K.T. thank Prof. Hiroaki Suzuki for fruitful discussions on the mixed potential.

## Author contributions
J.N. conceived the concept. J.N. and K.T. supervised the project. M.Y. and K.T. derived the equations. M.Y., K.T., and J.N. designed the figures. M.Y., N.A.P.N., J.N., and K.T. wrote the paper.

## Competing interests
The authors declare no competing interests.
