## [Peer Review File · Communications Chemistry]

Reviewers' comments:

Reviewer #1 (Remarks to the Author):

The authors have prepared a very detailed theoretical study on mixed potential catalysts. While this concept is not a wholly new concept, its exploration and widespread adoption has been limited. These new catalysts could potentially produce desired catalytic products without the input of external electrical energy based on the present materials half-reactions occurring at the electrode surface. This operates on the premise that there are two different materials in electrical contact where one shows a catalytic reaction exchange current that matches the exchange current of the other material at the open circuit potential (OCP) where no external current is flowing. The authors eventually relate this to the Gibbs free energy and heat of reaction or entropic losses generated by the reactions as joule heat ($\eta \cdot i$). The authors derive equations that rely on exchange current densities and are shown to affect the mixed potential. But I still would have liked to see more discussion and derivation of equations that included specific kinetic rate constants, especially when discussing catalytic reactions, where various reaction kinetics can significantly affect the observed currents. This is a well written paper and one that I think will be an important contribution in a new class of mixed potential or bifunctional catalysts where multiple reactions could be occurring at a given potential. There are a few points that I feel should be addressed outlined below. I recommend this paper be published with minor revisions.

Additional comments are included below.

1. Better explanation of the various electrode reactions and the defining the constants would help clarify the concepts. Equations 1 and 2 each occur on two different materials leading to i_{1I} , i_{2I} , i_{1II} , and i_{2II} . These constants should be defined as such in the text.

2. Reactions 1 and 2 each have a forward and backwards reaction arrow and thus, between the two equations, there are for possible electron transfer reaction directions, for each material. The authors only show the forward reactions in the Figure 2A schematic but the currents for the reverse reactions on each hypothetical material are plotted in 2B. Additionally, it seems the backwards reaction rates were included in equations 4 and 5. This needs to be clarified or fixed.

3. Equations 4 and 5 already have the mixed potential and equilibrium potentials included, but the mixed potential isn't defined beforehand. It is included in the schematic in Figure 2B with a brief description in the caption, but the mixed potential Greek symbol is not explicitly defined in the text. This should be changed to have the reaction overpotentials (Greek letter η), as shown in the derivation in the SI (equation S1-1).

4. Page 5 line 8, the authors state "substituting the equations of currents on component I and II into equation (6)". I am assuming this is referring to equations 4 and 5, but this should be clarified when deriving these equations.

5. The authors need to better explain the rearrangement of the equation S1-4 to equation S1-5 and why

the various exchange current densities disappear for the mixed potential terms so that it is clear to the general reader.

6.The authors state the assumption that $\alpha_1 = \alpha_2 = \alpha$, but the drawn currents in Figure 2B show that there is a difference in the charge transfer coefficients. Realistically, if one was to try to numerically model a mixed potential reaction rate at various conditions the value for α for the different anodic and cathodic reactions might need to be changed from one another. I think a better explanation of this is warranted for clarification.

7.Reactant concentration and mass transport also plays an important role in the half reaction current densities. While not necessary to derive here, the authors should put in some discussion about this and how it could affect the derived results.

8.The authors neglect to mention that the Gibbs free energy depends also on the number of electrons transferred in the reaction. The authors are clearly assuming the reactions defined in equations 4 and 5 are 1 electron transfer reactions. This should be explicitly stated in the text and discuss that for most of the equations $n = 1$ and it drops away – similar to the Gibbs free energy equation 19.

9.Page 9 lines 4-6, the authors state that by adjusting the exchange currents, researchers can control the current direction and thus enhance the selectivity for the desired products. This is a very optimistic statement. The authors should try to find a reference where the exchange current is changed (e.g. pH or concentration) to back up this statement as being plausible.

10.The authors reference many good references from Alan Bard including reference 34 but have missed another great reference from Alan Bard that would help support this work (ACS Analytical Chemistry - Anal. Chem. 2017, 89, 9843-9849, DOI: 10.1021/acs.analchem.7b01856). This study is similar in that the authors model the electrode half reactions (one set being catalytic and one set being from a simple redox molecule present in both oxidized and reduced forms) and could maybe even aid in answering comment 9 above.

11.Reference 36 has 2 periods at the end of it.

Reviewer #2 (Remarks to the Author):

This is an interesting study focusing on two electrochemical half reactions on a binary particle with different activities toward the half-reactions. The standard problem is two electrochemical half reactions on a unitary particle. So this work goes beyond the standard problem. In the new scenario, the dimensionality of the problem is four, while that of the standard problem is 2. Therefore, richer phenomena exist in this system. Overall, this reviewer is favorable of such conceptual/theoretical analysis, which is scarce nowadays and should be encouraged.

To further improve the quality of the manuscript and the clarity/depth of the conceptual/theoretical

analysis, I invite the authors to consider the following points.

(1) Could you give pragmatic examples of the binary particle-based catalytic systems? This will definitely enhance the relevance of the presented analysis.

(2) I notice a major flaw in the entropy part. The authors wrote, "in 23 the steady-state, the total entropy of the system dS remains constant at time dt ". This is untrue. Under steady-state conditions, dS/dt is a nonzero constant. We should distinguish between steady states and equilibrium states.

(3) The introduction section abounds with not-so-common concepts -- hurdles on the road toward proper understanding. For instance, "band-electron intermediates coupling". I suggest the authors adopt a more conservative description.

(4) The authors implied that their framework applies for both solid-gas and solid-liquid interfaces. Could you comment on the essential modifications of this framework when applied to these two kinds of interfaces, especially the later one.

(5) As regards the solid-liquid interfaces, the electric double layer (EDL) effects could be an interesting issue. The two components of the nanoparticle can possess different EDLs, and the two EDLs also communicate with each other. A recent study has touched upon this issue, see, <https://www.pnas.org/doi/10.1073/pnas.2307307120>. Though this reference studies a normal electrochemical cell, I see no essential difference as a normal electrochemical cell is a delocalized case of the binary particle considered here, namely, a case when components 1 and 2 are separated in space while connected via electrical wire. I believe, this is an interesting point to discuss, which will broaden the scope of this work from the special binary particle to any electrochemical cell.

Reviewers' comments:

Reviewer #1 (Remarks to the Author):

Comment 1-0:

The authors have prepared a very detailed theoretical study on mixed potential catalysts. While this concept is not a wholly new concept, its exploration and widespread adoption has been limited. These new catalysts could potentially produce desired catalytic products without the input of external electrical energy based on the present materials half-reactions occurring at the electrode surface. This operates on the premise that there are two different materials in electrical contact where one shows a catalytic reaction exchange current that matches the exchange current of the other material at the open circuit potential (OCP) where no external current is flowing. The authors eventually relate this to the Gibbs free energy and heat of reaction or entropic losses generated by the reactions as joule heat ($\eta \cdot i$). The authors derive equations that rely on exchange current densities and are shown to affect the mixed potential. But I still would have liked to see more discussion and derivation of equations that included specific kinetic rate constants, especially when discussing catalytic reactions, where various reaction kinetics can significantly affect the observed currents. This is a well written paper and one that I think will be an important contribution in a new class of mixed potential or bifunctional catalysts where multiple reactions could be occurring at a given potential. There are a few points that I feel should be address outlined below. I recommend this paper be published with minor revisions.

Response:

We thank the reviewer for the valuable comments. By addressing the constructive suggestions, we were able to strengthen the quality of our study further. We have responded to each comment individually as discussed below.

Comment 1-1:

1. Better explanation of the various electrode reactions and the defining the constants would help clarify the concepts. Equations 1 and 2 each occur on two different materials leading to i_{1I} , i_{2I} , i_{1II} , and i_{2II} . These constants should be defined as such in the text.

Response:

Thank you for your helpful comments. We have added the definitions of each current for equations 1 and 2 occurring on two different materials, I and II, as i_1^I , i_2^I , i_1^{II} , and i_2^{II} , in the revised manuscript.

Previous sentence:

“the reaction 1 and 2 can take place on each of catalyst components I and II”

Modified sentence (Page 4, line 10-11):

“Equations (1) and (2) each occur on two different catalyst components leading to i_1^I , i_2^I , i_1^{II} , and i_2^{II} .”

Comment 1-2:

2. Reactions 1 and 2 each have a forward and backwards reaction arrow and thus, between the two equations, there are four possible electron transfer reaction directions, for each material. The authors only show the forward reactions in the Figure 2A schematic but the currents for the reverse reactions on each hypothetical material are plotted in 2B. Additionally, it seems the backwards reaction rates were included in equations 4 and 5. This needs to be clarified or fixed.

Response:

Thank you for bringing this to our attention. The reviewer has correctly pointed out that both forward and backward reaction rates are involved in the Butler–Volmer equations 4 and 5. In response, we have added backward arrows to illustrate the reversible reactions in Figure 2A of the revised manuscript, which is consistent with the polarization curves shown in Figure 2B and the Butler–Volmer equations 4 and 5.

Previous Figure 2:

Modified Figure 2 (Page 4):

Fig. 2. A proposed mixed-potential-driven catalysis model. (A) Schematic of a mixed-potential-driven catalytic reaction occurring on the catalyst composed of component I and II. Cathodic and anodic half-reactions can occur in each of the component I and II. Electrons are transferred within and between the component I and II. (B) Illustration of the four polarization curves for the cathodic

and anodic half-reactions on catalyst component I and II. The mixed potential is the point at which the sum of the four currents is zero.

Comment 1-3:

3. Equations 4 and 5 already have the mixed potential and equilibrium potentials included, but the mixed potential isn't defined beforehand. It is included in the schematic in Figure 2B with a brief description in the caption, but the mixed potential Greek symbol is not explicitly defined in the text. This should be changed to have the reaction overpotentials (Greek letter eta), as shown in the derivation in the SI (equation S1-1).

Response:

Thank you for pointing out the important point. First, we have clearly defined the mixed potential ϕ^{mix} following the introduction of the Butler–Volmer equations (4)–(7) in the revised manuscript. Furthermore, the mixed potential term ϕ^{mix} in equations (4) and (5) of the previous manuscript has been changed to an arbitrary potential ϕ in the revised manuscript. Therefore, “ $\phi - \phi_1^{\text{eq}}$ ” and “ $\phi - \phi_2^{\text{eq}}$ ” are now designated as overpotentials “ η_1 ” and “ η_2 ”, respectively. Simultaneously, considering comments 1–4, we have additionally written the Butler–Volmer equations of Reaction 2 for components I and II in the revised manuscript.

Additionally, in the revised manuscript, the overpotentials upon the formation of the mixed potential are labelled as “ η_1^{mix} ” and “ η_2^{mix} ” as shown in the Equations (12) and (13). Therefore, all references to the overpotentials at mixed potential “ η_1 ” and “ η_2 ” in the previous manuscript have been updated to “ η_1^{mix} ” and “ η_2^{mix} ” throughout the revised manuscript, SI, and in Figures 1, 2, 3, 5, and S1.

Previous sentence:

“As shown in **Fig. 2B**, once a mixed potential is formed, the net current is zero according to the definition of the mixed potential.”

Modified sentence (Page 5, line 6-7):

“Here, the mixed potential ϕ^{mix} is defined as the potential at which the net current is zero as shown in **Fig. 2B**.”

Previous equations (Page 5):

$i_1^I = i_1^{I0} \left(e^{(1-\alpha_1)f(\phi^{\text{mix}} - \phi_1^{\text{eq}})} - e^{-\alpha_1 f(\phi^{\text{mix}} - \phi_1^{\text{eq}})} \right)$	(4)
$i_2^I = i_2^{I0} \left(e^{(1-\alpha_2)f(\phi^{\text{mix}} - \phi_2^{\text{eq}})} - e^{-\alpha_2 f(\phi^{\text{mix}} - \phi_2^{\text{eq}})} \right)$	(5)

Modified equations (Page 5):

Reaction 1 on component I: $i_1^I = i_1^{I0} \left(e^{(1-\alpha_1)f(\phi - \phi_1^{\text{eq}})} - e^{-\alpha_1 f(\phi - \phi_1^{\text{eq}})} \right)$	(4)
---	-----

Reaction 2 on component I: $i_2^I = i_2^{I0} \left(e^{(1-\alpha_2)f(\phi-\phi_2^{eq})} - e^{-\alpha_2f(\phi-\phi_2^{eq})} \right)$	(5)
Reaction 1 on component II: $i_1^{II} = i_1^{II0} \left(e^{(1-\alpha_1)f(\phi-\phi_1^{eq})} - e^{-\alpha_1f(\phi-\phi_1^{eq})} \right)$	(6)
Reaction 2 on component II: $i_2^{II} = i_2^{II0} \left(e^{(1-\alpha_2)f(\phi-\phi_2^{eq})} - e^{-\alpha_2f(\phi-\phi_2^{eq})} \right)$	(7)

Added sentence following equations (4)-(7) (Page 5, line 2-5):

“ i_1^{I0} , i_2^{I0} , i_1^{II0} , and i_2^{II0} are the exchange currents for reactions 1 and 2 on components I and II, respectively. $\phi - \phi_1^{eq}$ and $\phi - \phi_2^{eq}$ correspond to overpotentials η_1 and η_2 for reactions 1 and 2, respectively.”

Previous equations for the overpotentials at the mixed potential:

$ \eta_1 = \phi^{mix} - \phi_1^{eq} = \frac{1}{f} \ln \frac{e^{\alpha f(\phi_2^{eq} - \phi_1^{eq})} + \frac{(i_1^{I0} + i_1^{II0})}{(i_2^{I0} + i_2^{II0})}}{e^{-(1-\alpha)f(\phi_2^{eq} - \phi_1^{eq})} + \frac{(i_1^{I0} + i_1^{II0})}{(i_2^{I0} + i_2^{II0})}}$	(10)
$ \eta_2 = \phi_2^{eq} - \phi^{mix} = \frac{1}{f} \ln \frac{e^{(1-\alpha)f(\phi_2^{eq} - \phi_1^{eq})} + \frac{(i_2^{I0} + i_2^{II0})}{(i_1^{I0} + i_1^{II0})}}{e^{-\alpha f(\phi_2^{eq} - \phi_1^{eq})} + \frac{(i_2^{I0} + i_2^{II0})}{(i_1^{I0} + i_1^{II0})}}$	(11)

Modified equations for the overpotentials upon formation of the mixed potential:

$ \eta_1^{mix} = \phi^{mix} - \phi_1^{eq} = \frac{1}{f} \ln \frac{e^{\alpha f(\phi_2^{eq} - \phi_1^{eq})} + \frac{(i_1^{I0} + i_1^{II0})}{(i_2^{I0} + i_2^{II0})}}{e^{-(1-\alpha)f(\phi_2^{eq} - \phi_1^{eq})} + \frac{(i_1^{I0} + i_1^{II0})}{(i_2^{I0} + i_2^{II0})}}$	(12)
$ \eta_2^{mix} = \phi_2^{eq} - \phi^{mix} = \frac{1}{f} \ln \frac{e^{(1-\alpha)f(\phi_2^{eq} - \phi_1^{eq})} + \frac{(i_2^{I0} + i_2^{II0})}{(i_1^{I0} + i_1^{II0})}}{e^{-\alpha f(\phi_2^{eq} - \phi_1^{eq})} + \frac{(i_2^{I0} + i_2^{II0})}{(i_1^{I0} + i_1^{II0})}}$	(13)

Comment 1-4:

4. Page 5 line 8, the authors state “substituting the equations of currents on component I and II into equation (6)”. I am assuming this is referring to equations 4 and 5, but this should be clarified when deriving these equations.

Response:

Thank you for your suggestion. As mentioned in the response to comments 1–3, we have written the Butler–Volmer equations (4), (5), (6), and (7) explicitly for the four currents i_1^I , i_2^I , i_1^{II} , and i_2^{II} , respectively.

Previous sentence:

“Substituting the equations of currents on component I and II into Equation (6),”

Modified sentence (Page 5, line 8):

“By substituting the Equations (4), (5), (6) and (7) into Equation (8),”

Comment 1-5:

5. The authors need to better explain the rearrangement of the equation S1-4 to equation S1-5 and why the various exchange current densities disappear for the mixed potential terms so that it is clear to the general reader.

Response:

Thank you for your valuable comments. We have added an explanation for the rearrangement of Equations S1–4 to Equation S1–5, and then to S1–6, as follows.

Previous sentences (Supplementary information):

“which can be rearranged to:”

“Then, Eq. (S1-5) can be reduced to:”

Modified sentences (Supplementary information, Page 3, line 19-20, and 21-22):

“By extracting the exponential terms of mixed potential $e^{(1-\alpha)f\phi^{\text{mix}}}$ and $e^{-\alpha f\phi^{\text{mix}}}$, one can rearrange Eq. (S1-4) to Eq. (S1-5):”

“Then, by organizing $e^{(1-\alpha)f\phi^{\text{mix}}}$ terms and $e^{-\alpha f\phi^{\text{mix}}}$ terms to the left-hand and right-hand sides of the equation, respectively, Eq. (S1-5) can be reduced to:”

Comment 1-6:

6. The authors state the assumption that $\alpha_1 = \alpha_2 = \alpha$, but the drawn currents in Figure 2B show that there is a difference in the charge transfer coefficients. Realistically, if one was to try to numerically model a mixed potential reaction rate at various conditions the value for α for the different anodic and cathodic reactions might need to be changed from one another. I think a better explanation of this is warranted for clarification.

Response:

We appreciate the reviewer’s insightful question regarding the transfer coefficient and fully agree that the transfer coefficient also plays an important role in the position of the mixed potential. In Figure 2B, our intention is not to indicate the difference in the charge transfer coefficients, but to emphasize the effect of the exchange current on the polarization curves. Therefore, we have adjusted the activation overpotentials of i_1^{II} and i_2^I to demonstrate their low exchange currents of i_1^{II} and

i_2^1 . In the revised manuscript, we clearly show that the mixed potential can be numerically calculated by substituting Equations (4), (5), (6), and (7) into Equation (8) using the practical values of exchange currents, equilibrium potentials, and transfer coefficients. In addition, we modified the paragraph describing the analytical equations based on the assumptions for clarification.

Previous sentences (Page 5, line 9 and Page 6 line 19):

“Substituting the equations of currents on component I and II into Equation (6) and assuming $\alpha_1 = \alpha_2 = \alpha$, we can obtain a mixed potential of ϕ^{mix} at the steady-state of reaction as follows:”

“...two approximation methods were adopted. One is the Tafel approximation and the other is the linear approximation of the Taylor expansion...”

Added sentences (Page 5, line 8-13 and Page 6, line 20-23):

“By substituting Equations (4), (5), (6), and (7) into Equation (8), one can calculate the mixed potential ϕ^{mix} numerically using practical values of exchange currents, equilibrium potentials, and transfer coefficients. On the other hand, one can obtain the relationship among mixed potentials, overpotentials, and exchange currents based on analytical solutions with the assumption of identical transfer coefficients ($\alpha_1 = \alpha_2 = \alpha$). Then, one can derive Equation (9) for ϕ^{mix} (detailed derivation shown in **Supplementary Section 1**).”

“...two approximation methods were adopted. One is the linear approximation of the Taylor expansion for small overpotentials, and the other is Tafel approximation for large overpotentials (see **Supplementary Section 1** for the case of Tafel approximation).”

Comment 1-7:

7. Reactant concentration and mass transport also plays an important role in the half reaction current densities. While not necessary to derive here, the authors should put in some discussion about this and how it could affect the derived results.

Response:

We fully agree with the reviewer’s comment that the concentration and mass transport effects also play significant roles in half-reaction current densities. We have briefly discussed the effect of mass transport in the revised main text and Supplementary Section 7.

We consider an extreme case in which one reaction is solely charge-transfer-controlled and the other reaction is completely diffusion-limited. As shown in the figure below, the current in Reaction 2 is limited by mass transport rather than by activation kinetics, and mass transport determines the position of the mixed potential. We expect that, as the catalyst activity increases, situations arise in which mass transport effects should be considered as the rate-controlling factor.

Figure S5. Schematic plots of polarization curves for two half-reactions illustrate the mass transfer effect. Half-reaction 1 is solely charge-transfer controlled. The solid and dash lines of half-reaction 2 represent the currents of no mass transport effect and diffusion limiting, respectively. In this case, the mixed potential would shift to lower potentials from ϕ^{mix} to $\phi^{\text{mix}'}$.

In the revised manuscript, we have added discussion including mass transport as described below.

Added sentences (Page 12, line 30-39)

“The mass transport effect is not included in the present theoretical model because the main aim of this study was to show that catalytic activity mainly determines the mixed potential. However, it is necessary to consider the non-linear mass transport effect to determine the current value in addition to Butler-Volmer equations. The position of mixed potential and reaction rate are shifted depending on the mass transport effect, as discussed in **Supplementary Section 7**. Even more complex, electron transfer numbers, transfer coefficients, and the co-occurrence of thermal reactions must be considered in the kinetic model of mixed-potential-driven catalysis. In actual catalytic reaction systems, these additional effects must be considered in an extremely complex manner. Therefore, it is necessary to combine research on relatively simple systems to approach real catalytic reactions that involve extremely complex elements.”

Comment 1-8:

8. The authors neglect to mention that the Gibbs free energy depends also on the number of electrons transferred in the reaction. The authors are clearly assuming the reactions defined in equations 4 and 5 are 1 electron transfer reactions. This should be explicitly stated in the text and discuss that for most of the equations $n = 1$ and it drops away – similar to the Gibbs free energy equation 19.

Response:

Thank you for your comments. In the revised manuscript, we explicitly state that the theoretical model corresponds to the case where $n = 1$. This simplicity facilitates a clear understanding of how the free-energy drop is converted into overpotentials.

In addition, we have added a discussion for $n \geq 2$, in which the determination of overpotentials is not easy. The partitioning of the overpotential across multiple electron-transfer reactions depends on the specific kinetics of each elementary electron-transfer step. For example, for $n = 2$, the Gibbs free energy drop of the overall reaction is not always converted to overpotential by 50%, as shown in Equations (S5–7). If one of the electron transfer steps is rate-limiting, the Gibbs free energy drop is almost completely converted into the overpotential of that particular step. As shown in Equations (S5–7), n is not always equal to n' , where the first term of $nF(\phi_2^{\text{eq}} - \phi_1^{\text{eq}})$ is the driving force of the net reaction, and the later term is corresponding to the overpotential energy.

$$-\Delta G_r = nF(\phi_2^{\text{eq}} - \phi_1^{\text{eq}}) = n'F(\phi_2^{\text{eq}} - \phi^{\text{mix}} + \phi^{\text{mix}} - \phi_1^{\text{ocp}}) = n'F(|\eta_1| + |\eta_2|) \quad (\text{S5-7})$$

However, we assumed $n = n' = 1$ in the main text for simplicity to obtain Equation (21). Therefore, we have added the above equation and corresponding discussion in the revised manuscript.

Previous sentences:

Page 4, line 4: “where anodic reaction 1 and cathodic reaction 2 can occur in both components I and II of the catalyst”

Modified sentences:

Page 4, line 4: “where we assume one-electron transfer processes of anodic reaction 1 and cathodic reaction 2 occurring at both components I and II of the catalyst.”

Added sentences:

Page 8, line 21-13: “We assumed that half-reactions 1 and 2 are one-electron transfer reactions so that the number of “moles of electrons” exchanged in the half-reactions, n , is equal to 1 and drops away in Equation (21).”

Page 7 of the supplementary information: we have added n in Eqs. (S2-1a)-(S2-1d) and “and n is the stoichiometric number of electrons consumed in the electrode reaction (in our case, $n = 1$)”.

Page 11 of the supplementary information: we added n in Eqs. (S3-21).

Page 14 of the supplementary information: we added “where n' is the electron transfer number in the rate limiting step, respectively. We assumed $n = n' = 1$ in the main text for simplicity so that we get Eq. (21).” following Eq. (S5-7).

Page 14 of the supplementary information: we added “where $n = 1$ in our case” after Eq. (S5-8).

Comment 1-9:

9. Page 9 lines 4-6, the authors state that by adjusting the exchange currents, researchers can control the current direction and thus enhance the selectivity for the desired products. This is a very optimistic statement. The authors should try to find a reference where the exchange current is changed (e.g. pH or concentration) to back up this statement as being plausible.

Response:

Thank you for your helpful comments. Following the reviewer’s comments, we have added to the revised manuscript that the exchange current changes depending on the concentration of redox molecules and pH. In addition, we have added the reference A. Bard, ACS Analytical Chemistry - Anal. Chem. 2017, 89, 9843–9849, DOI: 10.1021/acs.analchem.7b01856, as described below.

Added sentences (Page 9, line 24-26):

“The exchange current values are sensitive to substance concentrations and pH, as reported in the literature.⁴⁷ Controlling the current direction by adjusting the exchange current can help researchers harness the benefits of the internal electric field of the catalyst and enhance selectivity for the desired products.”

Comment 1-10:

10. The authors reference many good references from Alan Bard including reference 34 but have missed another great reference from Alan Bard that would help support this work (ACS Analytical Chemistry - Anal. Chem. 2017, 89, 9843-9849, DOI: 10.1021/acs.analchem.7b01856). This study is similar in that the authors model the electrode half reactions (one set being catalytic and one set being from a simple redox molecule present in both oxidized and reduced forms) and could maybe even aid in answering comment 9 above.

Response:

Thank you for introducing Professor Bard's excellent paper. We have cited this paper in response to comment 1-9.

Comment 1-11:

11. Reference 36 has 2 periods at the end of it.

Response:

Thank you for pointing it out.

Reviewer #2 (Remarks to the Author):

Comment 2-0:

This is an interesting study focusing on two electrochemical half reactions on a binary particle with different activities toward the half-reactions. The standard problem is two electrochemical half reactions on a unitary particle. So this work goes beyond the standard problem. In the new scenario, the dimensionality of the problem is four, while that of the standard problem is 2. Therefore, richer phenomena exist in this system. Overall, this reviewer is favorable of such conceptual/theoretical analysis, which is scarce nowadays and should be encouraged.

To further improve the quality of the manuscript and the clarity/depth of the conceptual/theoretical analysis, I invite the authors to consider the following points.

Response:

We thank the reviewer for the encouraging and valuable comments, which have improved the quality of the manuscript. By addressing the constructive criticism, we were able to strengthen the

quality and impact of our study further. Each point is addressed below with a description of the actions taken upon revision.

Comment 2-1:

(1) Could you give pragmatic examples of the binary particle-based catalytic systems? This will definitely enhance the relevance of the presented analysis.

Response:

Hutchings et al. reported a binary particle-based catalytic system for hydroxymethylfurfural oxidation over Au–Pd catalysts. The thermal oxidation of hydroxymethylfurfural over bimetallic physical mixtures was well described by considering the electrochemical coupling of the two half-reactions. [Huang, X. et al. *Au–Pd separation enhances bimetallic catalysis of alcohol oxidation*. *Nature* **603**, 271–275 (2022); Daniel, I. T. et al. *Electrochemical Polarization of Disparate Catalytic Sites Drives Thermochemical Rate Enhancement*. *ACS Catal.* **13**, 14189–14198 (2023).]

Moreover, we introduced the production of ethanol during the hydrogenation of CO₂ as an unexpected product in thermal reactions in which binary CuPd powder catalysts are used in the presence of water. We studied the mechanism from the viewpoint of mixed-potential-driven catalysis using a model reactor that included Cu and Pd electrodes in a CO₂/H₂ mixture gas and water. A current was detected between the short-circuited Cu and Pd electrodes, indicating that CO₂ reduction and H₂ oxidation occurred simultaneously over the Cu and Pd catalysts. [Takeyasu, K. et al. *Experimental Verification of Mixed-potential-driven Catalysis*. *e-Journal Surf. Sci. Nanotechnol.* **21**, 164–168 (2022).].

A brief description of pragmatic examples has been added to the revised manuscript.

Previous sentences:

“A similar mechanism, known as cooperative redox enhancement, has been reported for the oxidation of alcohols (hydroxymethylfurfural) on Au-based catalysts.^{12,13,16} The occurrence of a mixed-potential-driven reaction during the hydrogenation of 4-nitrophenol¹⁵ and CO₂¹⁴ were also proposed previously. Unlike conventional thermal catalysis, it produces different products without the need for external energy. These findings indicate that mixed-potential-driven catalysis represents a new category of heterogeneous catalysis. Because the mixed-potential-driven catalysis fundamentally changes the activity and selectivity of catalysts, it is expected to be applied to industrial catalysts in the future.”

Modified and added sentences (Page 2, line 10-11 and 20-28):

“Here, we introduce the concept of “mixed-potential-driven catalysis” as such catalytic systems.”

“More interestingly, the mixed-potential-driven mechanism is caused by binary heterogeneous catalysts. The oxidation of alcohols (hydroxymethylfurfural) on Au-Pd binary catalysts seems to proceed via mixed-potential-driven catalysis.^{12,13,16} It is also worth noting that ethanol is produced with surprisingly high selectivity by CO₂ hydrogenation on CuPd binary powder catalysts in the presence of water, which is an unexpected product in thermal catalysis.¹⁴ These reports strongly suggest that electrochemical processes play a role in controlling the activity and selectivity of heterogeneous catalysis without the need for external energy. Mixed-potential-driven catalysis is expected to open up a new category of heterogeneous catalysis in both basic research and industrial applications.”

Here, we would like to introduce our current study that demonstrates mixed-potential-driven catalysis. One paper related to glucose oxidation was submitted to a journal. In addition, we are currently preparing another manuscript related to CO oxidation on Au, TiO₂, and N-doped-reduced graphene oxide (NrGO) catalysts. In these experiments, we separated the two catalysts in the same solution and short-circuited them. We successfully measured the short-circuit current without applying external energy. Moreover, we observed a strong correspondence between the measured mixed potentials from the short-circuit experiments and the predicted mixed potentials from the polarization curves. Based on these findings, we are confident of our understanding of the mixed-potential-driven catalytic mechanism.

We de-coupled the glucose oxidation (anodic) and oxygen reduction (cathodic) reactions to examine the mixed-potential-driven reaction mechanism in glucose oxidation. Au NPs supported on activated carbon (Au/C) acted as macroscopic anodes, whereas Pt/C, Pd/C, and NrGO acted as macroscopic cathodes in an identical environment. This finding allowed us to monitor the short-circuit current without external energy, which is a direct indicator of mixed-potential-driven catalysis. We successfully measured the short-circuit current of Au/C (working electrode) with Pt/C (counter electrode) (Figure R1). We then recorded the polarization curves of the Au/C and Pt/C catalysts for the glucose oxidation half-reaction (GOR) and oxygen reduction half-reaction (ORR). The GOR curve was collected with 0.1 M glucose in the absence of O₂, whereas the ORR curve was obtained without glucose under O₂-saturated conditions. The mixed potential predicted from the polarization curves was the point at which the sum of the four currents was zero. The agreement between the predicted and measured mixed potentials indicated that mixed-potential-driven reaction mechanisms were operative in the glucose oxidation system.

Fig. R1. Verification theoretical framework of mixed-potential-driven catalysis in glucose oxidation on bicomponent catalysts. (a) Short-circuit current (top) and measured mixed potential (bottom) as a function of time for the Au/C (working electrode) with Pt/C (counter electrode). (b) Polarization curves for the Au/C and Pt/C catalysed the GOR and ORR.

Similarly, if the mixed-potential-driven reaction mechanisms are valid for CO oxidation ($\text{CO} + \frac{1}{2}\text{O}_2 \rightarrow \text{CO}_2$), the electrons generated from the CO oxidation half-reactions are consumed by the O₂ reduction half-reactions. As shown in Figure R2a, we used a simple single-cell setup. AuNPs and NrGO were prepared as individual catalysts. The two electrodes were externally short circuited. Without any externally applied potential, positive short-circuited currents are detected when CO and CO + O₂ are flown, as shown in Figure R2b. Therefore, it can be concluded that a mixed-potential-

driven reaction mechanism exists for CO oxidation.

Fig. R2. Verification theoretical framework of mixed-potential-driven catalysis in CO oxidation. (A) Schematic representation of the reaction cell used to measure the short-circuited current. There is no external applied potential and both the Au NPs and NrGO electrodes are under identical gas/liquid environments. (B) Short-circuited current at a 2-hour periodic switch between CO (1.5%) and CO (1.125%) + Ar (73.875%) + O₂ (25%) for the Au NPs (working electrode) and NrGO (counter electrode) in the single cell.

Comment 2-2:

(2) I notice a major flaw in the entropy part. The authors wrote, "in 23 the steady-state, the total entropy of the system dS remains constant at time dt ". This is untrue. Under steady-state conditions, dS/dt is a nonzero constant. We should distinguish between steady states and equilibrium states.

Response:

Thank you very much for your comment on non-equilibrium thermodynamics. We carefully checked several textbooks and papers regarding d_iS_{sys} , the entropy production by the irreversible processes inside the system. d_iS_{sys} is known as “uncompensated transformation”, which appeared in non-equilibrium thermodynamics. Probably, there is some misunderstanding regarding the definition of the “total entropy of the system dS ”. We guess that the reviewer thought dS is the total entropy of the system and environment. Therefore, in the revised manuscript, we have explicitly defined entropy as the system entropy change dS_{sys} , the environment entropy change dS_{env} , and the total entropy change of the system and environment dS_{tot} . The relationship of the entropy change is given as follows:

$$dS_{tot} = dS_{sys} + dS_{env} \quad (1)$$

Notably, the system entropy change dS_{sys} , is divided into d_eS_{sys} and d_iS_{sys} :

$$dS_{sys} = d_eS_{sys} + d_iS_{sys} \quad (2)$$

where d_eS_{sys} is the entropy change due to exchange of matter and energy with the exterior and d_iS_{sys} is the entropy change due the “uncompensated transformation,” the entropy produced by the irreversible processes in the interior of the system, which is provided in the textbook by Prigogine [Kondepudi, D. & Prigogine, I. *Modern Thermodynamics: From Heat Engines to Dissipative Structures*. (John Wiley & Sons, 1998).].

Page 88:

We may begin the modern formalism by expressing the changes in entropy as a sum of two parts [17]:

$$dS = d_eS + d_iS \quad (3.4.5)$$

in which d_eS is the entropy change due to exchange of matter and energy with the exterior and d_iS is the entropy change due to “uncompensated transformation,” the entropy produced by the irreversible processes in the interior of the system (Fig. 3.7).

It should be noted here that d_iS_{sys} corresponds to the driving force term for chemical reactions. The following equation is obtained from the textbook written by Prigogine.

Page 388:

In a stationary state the total entropy of the system remains constant, i.e.

$$\frac{dS}{dt} = \frac{d_eS}{dt} + \frac{d_iS}{dt} = 0 \quad \text{where} \quad \frac{d_iS}{dt} = \int_V \sigma dV > 0 \quad (17.1.12)$$

which means the entropy exchange with the exterior must be negative:

$$\frac{d_eS}{dt} = -\frac{d_iS}{dt} < 0 \quad (17.1.13)$$

The abovementioned equations are obtained from the literature showing d_iS_{sys} . In the steady-state, S_{sys} is constant, and all entropy generated is continuously released into the environment [Tomé, T. & De Oliveira, M. J. *Entropy production in nonequilibrium systems at stationary states. Phys. Rev. Lett.* **108**, 1–5 (2012).]. Therefore, the time derivative of Equation (2) is zero [Zhang, X. J., Qian, H. & Qian, M. *Stochastic theory of nonequilibrium steady states and its applications. Part I. Phys. Rep.* **510**, 1–86 (2012). Page 65; Malouf, W. T. B., Santos, J. P., Correa, L. A., Paternostro, M. & Landi, G. T. *Wigner entropy production and heat transport in linear quantum lattices. Phys. Rev. A* **99**, 52104 (2019).]:

$$\frac{dS_{\text{sys}}}{dt} = \frac{d_eS_{\text{sys}}}{dt} + \frac{d_iS_{\text{sys}}}{dt} = 0 \quad \text{or} \quad \frac{d_iS_{\text{sys}}}{dt} = -\frac{d_eS_{\text{sys}}}{dt} \quad (3)$$

There is no irreversible process in the environment, resulting in the following equation:

$$\frac{dS_{\text{env}}}{dt} = \frac{d_eS_{\text{env}}}{dt} \quad (4)$$

Since the environment entropy change dS_{env} is solely attributed to the exchange with the system entropy change, we can obtain

$$\frac{dS_{\text{env}}}{dt} = \frac{d_eS_{\text{env}}}{dt} = -\frac{d_eS_{\text{sys}}}{dt} \quad (5)$$

Finally, at steady-state, the total entropy change rates of the system and environment are equal to the entropy production rate in the system [Yan, H., Zhang, F. & Wang, J. *Thermodynamic and dynamical predictions for bifurcations and non-equilibrium phase transitions. Commun. Phys.* **6**, (2023).]:

$$\frac{dS_{\text{tot}}}{dt} = \frac{dS_{\text{env}}}{dt} = \frac{d_i S_{\text{sys}}}{dt} \quad (6)$$

Therefore, dS_{tot}/dt is a nonzero constant at steady-state conditions.

Briefly, in the steady-state, dS_{tot}/dt is a nonzero constant; however, dS_{sys}/dt is equal to zero. In the equilibrium state, $d_i S_{\text{sys}}/dt = 0$, then $dS_{\text{tot}}/dt = 0$. In our original manuscript, we simply wrote dS , which may have caused confusion. In the revised manuscript, we have clearly defined the system entropy change as dS_{sys} to avoid any misunderstanding in the text and Figure 5.

Modified symbol:

$dS \rightarrow dS_{\text{sys}}$; $d_i S \rightarrow d_i S_{\text{sys}}$; $d_e S \rightarrow d_e S_{\text{sys}}$

Added sentences (Page 10, line 16-18):

“The entropy production term $d_i S_{\text{sys}}$, can serve as a basis for the systematic description of irreversible processes occurring in a system, and $d_i S_{\text{sys}}$ is always non-negative. Moreover, in the steady-state, the time derivative of the system entropy, dS_{sys}/dt , is zero, that is, the entropy spontaneously generated inside the system is balanced by a flow of the entropy exchange with outside.”

Comment 2-3:

(3) The introduction section abounds with not-so-common concepts -- hurdles on the road toward proper understanding. For instance, "band-electron intermediates coupling". I suggest the authors adopt a more conservative description.

Response:

We agree with the reviewer’s comment. In the revised manuscript, we have removed the words “band-electron intermediates coupling” and “cooperative redox enhancement.”

We have deleted the sentence: “In other words, the energies of the band-electron intermediates coupling the two half-reactions are affected by changes in the electrochemical potential of the catalyst, which sets the relative driving force of each half-reaction.¹¹”

Previous sentence:

“A similar mechanism, known as cooperative redox enhancement, has been reported for the oxidation of alcohols (hydroxymethylfurfural) on Au-based catalysts.”

Modified sentence (Page 2, line 21-22):

“The oxidation of alcohols (hydroxymethylfurfural) on Au-Pd binary catalysts seems to proceed via mixed-potential-driven catalysis.^{12,13,16}”

Comment 2-4:

(4) The authors implied that their framework applies for both solid-gas and solid-liquid interfaces. Could you comment on the essential modifications of this framework when applied to these two kinds of interfaces, especially the later one.

Response:

We thank the reviewer for the insightful comments. As the reviewer points out, our theoretical framework can be applied to both solid–gas and solid–liquid interfaces. The point of the theoretical framework is based on the fact that the total free energy drop of the reaction consisting of the anodic and cathodic reactions is converted into overpotentials applied to each electrode; however, how the overpotential causes the reaction to proceed depends on the electric double layer (EDL) on the electrode surfaces. The essential difference between solid and solid–gas and solid–liquid interfaces is the local structure of the EDL formed depending on the overpotential, where electrochemical reactions are accelerated or decelerated. In this study, we would like to describe our speculations. The nanoscale EDL at the interface plays a significant role, and the shape of the local electric field of the EDL is determined by the concentration and distribution of cations, anions, and electrons depending on the overpotential. The local electric field critically influences the reaction kinetics. Therefore, it is important to study the local structure of the EDL at gas–solid and liquid–solid interfaces. Furthermore, as the electrode size decreases, a strong electric field is generated. To answer the reviewer’s question of “essential modifications,” it is necessary to clarify the relationship among the overpotential, electrode structure, and EDL structure. Even more complex mass transport effects, electron transfer numbers, and co-occurrence of thermal reactions must be considered. In actual catalytic systems, these effects are extremely complex. Therefore, we believe that research should be conducted using relatively simple modeling systems.

Added sentences (Page 12, line 15-29):

“In principle, the theoretical framework of mixed-potential-driven catalysis can be applied to both solid-gas and solid-liquid interfaces, where an electrolyte is necessary to convey ions. One open issue is how the overpotential is applied to electrode reactions at solid-gas and solid-liquid interfaces. At present, we consider that the overpotential in mixed-potential-driven catalysis corresponds to an electric double layer (EDL) at the catalyst surface, where electrochemical reactions are accelerated or decelerated. The nanoscale EDL at the interface may play a large role, where the shape of local electric field of EDL is determined by concentrations and distributions of cations, anions, and electrons depending on the overpotential. That local electric field should critically influence the reaction kinetics. Therefore, it is important to study the local structure of the EDL at the gas-solid and liquid-solid interfaces. Furthermore, as the size of the electrode decreases, a strong electric field may be generated. Thus, it is necessary to clarify the relationship among the overpotential, electrode structure, and EDL structure. Recent studies have reported that EDLs at spatially distant cathodes and anodes change in an intrinsically coupled manner.⁵¹ Future research will employ both experimental and theoretical studies of the EDL in mixed-potential-driven catalysis.”

Inspired by the reviewer’s thoughtful comment, the term “solution” has been changed to “electrolyte” in the revised Fig. 3.

Previous Figure 3:

Modified Figure 3:

Fig. 3. A conceptual electric series circuit illustrates a mixed-potential-driven catalytic reaction.

Comment 2-5:

(5) As regards the solid-liquid interfaces, the electric double layer (EDL) effects could be an interesting issue. The two components of the nanoparticle can possess different EDLs, and the two EDLs also communicate with each other. A recent study has touched upon this issue, see, <https://www.pnas.org/doi/10.1073/pnas.2307307120>. Though this reference studies a normal electrochemical cell, I see no essential difference as a normal electrochemical cell is a delocalized case of the binary particle considered here, namely, a case when components 1 and 2 are separated in space while connected via electrical wire. I believe, this is an interesting point to discuss, which will broaden the scope of this work from the special binary particle to any electrochemical cell.

Response:

We completely agree with this comment regarding EDL, as written in the response to comment 2-4. The paper reported in PNAS was cited to emphasize the relationship between the overpotentials and EDL in the revised manuscript. We also believe that there is no essential difference between normal electrochemical cells and binary particle catalysts. However, the involvement of the electrochemical mechanisms in heterogeneous catalysis has not yet been sufficiently investigated. The present theoretical framework is fundamental and was constructed using a simple model. Therefore, numerous additional effects must be considered for further development and application. The EDL structure, mass transport effects, electron numbers, and co-occurrence of the thermal reactions should be considered. Based on your comments, we have substantially revised the

discussion section to describe our scope, including non-equilibrium thermodynamics and biological systems.

Added sentences (Page 12, line 15-Page13, line 2):

“In principle, the theoretical framework of mixed-potential-driven catalysis can be applied to both solid-gas and solid-liquid interfaces, where an electrolyte is necessary to convey ions. One open issue is how the overpotential is applied to electrode reactions at solid-gas and solid-liquid interfaces. At present, we consider that the overpotential in mixed-potential-driven catalysis corresponds to an electric double layer (EDL) at the catalyst surface, where electrochemical reactions are accelerated or decelerated. The nanoscale EDL at the interface may play a large role, where the shape of local electric field of EDL is determined by concentrations and distributions of cations, anions, and electrons depending on the overpotential. That local electric field should critically influence the reaction kinetics. Therefore, it is important to study the local structure of the EDL at the gas-solid and liquid-solid interfaces. Furthermore, as the size of the electrode decreases, a strong electric field may be generated. Thus, it is necessary to clarify the relationship among the overpotential, electrode structure, and EDL structure. Recent studies have reported that EDLs at spatially distant cathodes and anodes change in an intrinsically coupled manner.⁵¹ Future research will employ both experimental and theoretical studies of the EDL in mixed-potential-driven catalysis.

The mass transport effect is not included in the present theoretical model because the main aim of this study was to show that catalytic activity mainly determines the mixed potential. However, it is necessary to consider the non-linear mass transport effect to determine the current value in addition to Butler-Volmer equations. The position of mixed potential and reaction rate are shifted depending on the mass transport effect, as discussed in **Supplementary Section 7**. Even more complex, electron transfer numbers, transfer coefficients, and the co-occurrence of thermal reactions must be considered in the kinetic model of mixed-potential-driven catalysis. In actual catalytic reaction systems, these additional effects must be considered in an extremely complex manner. Therefore, it is necessary to combine research on relatively simple systems to approach real catalytic reactions that involve extremely complex elements.

Another important aspect of mixed-potential-driven catalysis is non-equilibrium thermodynamics. Mixed potential-driven catalysis will be particularly important in the energetics of enzymatic reactions of biological systems. As described above, the Gibbs free energy drop or uncompensated heat ($d_i S_{\text{sys}}$) is first converted into overpotential and then into heat. This energy conversion is a characteristic feature of non-equilibrium thermodynamics and is expected to greatly contribute to the future development of non-equilibrium thermodynamics itself. The energy conversion is particularly important in enzymatic reactions of biological systems is important because the mechanism of thermogenesis in biological systems is expected to be closely linked to the present non-equilibrium theory.⁵²”

REVIEWERS' COMMENTS:

Reviewer #1 (Remarks to the Author):

The authors have done a great job in responding to my points and alleviating my concerns. I recommend this paper be published without further edits.

Reviewer #2 (Remarks to the Author):

The authors have made excellent efforts to address my previous comments and I am now recommending publication.

Reviewer #1 (Remarks to the Author):

The authors have done a great job in responding to my points and alleviating my concerns. I recommend this paper be published without further edits.

Response:

We appreciate the reviewer for the positive feedback and the thorough review of our manuscript.

Reviewer #2 (Remarks to the Author):

The authors have made excellent efforts to address my previous comments and I am now recommending publication.

Response:

We express our gratitude to the reviewer for the insightful comments and constructive feedback.